# A Neural-Network-Based Landscape Search Engine: LSE Wisconsin

Matthew Haffner [1,*] , Matthew DeWitte [1], Papia F. Rozario [1] and Gustavo A. Ovando-Montejo [2]

1  Department of Geography and Anthropology, University of Wisconsin–Eau Claire, Eau Claire, WI 54701, USA; rozaripf@uwec.edu (P.F.R.)
2  Department of Environment and Society, Utah State University, Blanding, UT 84511, USA
*  Correspondence: haffnerm@uwec.edu; Tel.: +1-715-836-2316

**Featured Application: This paper introduces a neural-network-based landscape search engine tool for the state of Wisconsin. It provides several examples of how the application works and suggests avenues for future research.**

**Abstract:** The task of image retrieval is common in the world of data science and deep learning, but it has received less attention in the field of remote sensing. The authors seek to fill this gap in research through the presentation of a web-based landscape search engine for the US state of Wisconsin. The application allows users to select a location on the map and to find similar locations based on terrain and vegetation characteristics. It utilizes three neural network models—VGG16, ResNet-50, and NasNet—on digital elevation model data, and uses the NDVI mean and standard deviation for comparing vegetation data. The results indicate that VGG16 and ResNet50 generally return more favorable results, and the tool appears to be an important first step toward building a more robust, multi-input, high resolution landscape search engine in the future. The tool, called LSE Wisconsin, is hosted publicly on ShinyApps.io

**Keywords:** image retrieval; remote sensing; web GIS; GIScience





## 1. Introduction

Deep learning (DL) has been extensively and successfully applied in the field of remote sensing for tasks such as object detection, object segmentation, and land use classification [1]. Such methods have brought about major advancements in the discipline and have been crucial to the fusion of data science and remote sensing. At the same time, however, image retrieval—that is, returning similar images given a single input image—has become an increasingly common data science task, yet its application to remotely sensed datasets has been lacking. This project seeks to fill that gap in research through the creation of a "landscape search engine" tool, designed particularly for (though certainly not limited to) location analysis applications.

To achieve this goal, the authors leverage several common DL models—VGG16, ResNet-50, and NasNet—on digital elevation model (DEM) data and combine these outputs with a traditional vegetation metric, the normalized difference vegetation index (NDVI), in creating the image retrieval tool. The authors present this as a publicly accessible web application (https://uwec-geog.shinyapps.io/lse-wi accessed on 6 August 2023) which allows users to retrieve similar landscapes in the US state of Wisconsin for a location they select on the map. Using sliders and drop-down list options, users can select a specific neural network (NN) model, the number of locations to retrieve, the relative weight on terrain or vegetation, the amount of weight to place on mean vs. standard deviation NDVI, and an optional exclusion radius from the input location. To date, this is the only landscape search tool built specifically for the state of Wisconsin, and, to the authors' knowledge, it is the only search engine tool which leverages neural network models for landscape

search. Considering the increasing impact of data science on the domains of geographic information science (GIScience) and remote sensing, the development of this tool and its corresponding metrics signifies a crucial stride towards the creation of robust, user-friendly digital resources for the research community and end-users alike.

*Background*

Implementations of DL in remote sensing and within the broader field of geographic information science (GIScience) have been applied to a variety of tasks, such as land cover mapping [2], environmental parameter retrieval [3], data fusion and downscaling [4], object detection [5], and information construction and prediction (see [1,6,7] for comprehensive overviews). Other efforts have focused on advancing the principles of DL in remote sensing, including the integration of aerial images, and the detection of small objects on the landscape [5,8]. Yuan et al. [3], in particular, have advocated for the fusion of geographic principles into DL for remote sensing tasks, most notably Tobler's famous First Law of Geography: "Everything is related to everything else, but near things are more related than distant things". The most common and mainstream frameworks are back-propagation NNs, such as convolutional neural networks (CNNs). Indicative of their power, CNN models often produce a sizable increase in accuracy over traditional regression models, particularly when working with remotely sensed data. Further, unlike traditional learning algorithms, intrinsic features from raw input data can be extracted using a variety of DL frameworks without using manual digitizing techniques, thus reducing the need for reliance on domain knowledge [9].

Despite the significant number of remote sensing studies which utilize DL, there is a paucity of research on the particular task of image retrieval using remotely sensed data, with a few notable exceptions. Jasiewicz et al. [10] first coined the term "landscape search engine" in building a landscape similarity tool for terrain across the entire country of Poland. Using the concept of "geomorphons", this approach classifies pixels from digital elevation models (DEMs) into several types: ridge, shoulder, spur, slope, hollow, footslope, valley, pit, flat, and peak. Another landscape similarity tool, developed by Dilts et al. [11], has been applied toward location optimization of control sites based on the spatial characteristics of treatment sites. The researchers applied a moving window analysis to generate per-pixel maps of similarity between the treatment and control areas for site selections. Outside of this application, the United States Geological Survey (USGS) has a landscape search tool focusing on land treatment exploration within the United States, making use of modifiable parameters, such as soil and vegetation characteristics [12]. Through an interactive web map, it allows users to input empirical characteristics for the purpose of finding areas with similar heat load, soil properties, and climate conditions. At the time of writing, however, the two formerly mentioned studies do not have publicly available toolkits, and none of these prior implementations make use of NNs.

VGG16, ResNet-50, and the Neural Architecture Search Network (NasNet) have been used frequently in remote sensing. The Visual Geometry Group (VGG) model architecture is a standard CNN which uses a specified number of consecutive convolutional layers to extract features from image data. The input of VGG is an image with resolution 224 × 224, and, since VGGNet is a classification network, the output shape is proportional to the number of classes in the dataset. The model architecture consists of multiple convolution layers followed by max pooling layers, and the end of the model consists of fully connected layers followed by the final classification layer. Two common VGG architectures used are VGG16 and VGG19, which are sixteen and nineteen layers deep, respectively [13]. The VGG16 architecture, in particular, was first introduced by Simonyan and Zisserman [14] for image recognition and has been used extensively in multispectral and hyperspectral image classifications even with low resolution imagery [15]. It has also been utilized for tasks such as road feature extraction [16], sea ice classification [17], image stitching [18], and many others.

ResNet-50 falls into the family of deep residual networks and contains 50 layers: 48 convolutional layers, one average pooling layer, and one max pooling layer [19]. This model, along with small modifications to its architecture, has been successfully applied in many remote sensing applications, such as image segmentation [20], classification [21,22], and image captioning [23]. In a comparative study of several NN models for remote sensing classification, ResNet-50 indeed outperformed other models, including NasNet and VGG16 [24]. NasNet has been applied to tasks such as scene classification (e.g., see [24,25]) but has been used for remote sensing tasks less often than VGG16 and ResNet-50. This makes its use in new applications of particular interest as a comparison with more commonly utilized models.

It should be noted that the issues associated with image retrieval for landscapes vary markedly from those associated with image retrieval on traditional color photographs. Whereas a picture of a red ball against the backdrop of green grass and a blue sky exhibits stark within-image pixel differences (i.e., high contrast), the continuous nature of the Earth's surface makes such extreme differences uncommon in landscape qualities like elevation. Similarly, the variability of color in a natural landscape is much less than what is present in photos containing human objects, such as vehicles and clothing. For these reasons, it is worth exploring the utility of DL for image retrieval with landscape data.

## 2. Methods and Data

Due to the often long computation times incurred by using NN models and in making vector geometry calculations, the code used to create LSE Wisconsin was grouped into three stages: (a) data extraction, (b) a priori modeling, (c) and ad hoc querying. We notably take a different approach from Jasiewicz et al. [10] by using NNs rather than geomorphons, additionally utilizing vegetation data, and allowing users to select a variety of model options. Further, our work is differentiated by the fact that the models make no explicit classification of pixels into various terrain types. In addition to taking advantage of state-of-the-art algorithms, this approach adds the benefit of flexibility.

### 2.1. Data Extraction

Two freely available remotely sensed data sources were utilized in this project: DEM data and NDVI data (see Figure 1). The DEM data comes from the Wisconsin Department of Natural Resources (DNR), and a 30m DEM resolution was selected to produce reasonable computation times given the size of the state of Wisconsin. This data is available for direct download as a single file from the Wisconsin DNR. Using a command line utility from the Geospatial Data Abstraction Library (GDAL), this single file was retiled into individual .tif files, each $256 \times 256$ pixels. Thus, the resulting extent of each .tif was about 7.5 km $\times$ 7.5 km, which resulted in a total of 2510 observations after removing .tif files which were completely empty (i.e., those at Wisconsin's borders). This size balances ease of computation while keeping a user-friendly approach. Medium-sized cities such as Eau Claire and La Crosse can mostly be covered by 1–2 grid cells, whereas larger cities such as Madison and Milwaukee are encompassed by more cells. It also strikes a reasonable balance between substantial terrain and vegetation variation between grid cells on the application, without burdening users with an overwhelming number (i.e., tens of thousands) of small grid cells as selection options.

The vegetation data comes from the National Air and Space Administration's (NASA) Moderate Resolution Imaging Spectroradiometer (MODIS) program, specifically, the 16-Day L3 Global 250 m SIN Grid. Similar to the DEM data, this dataset is available as a single HDF5 file. Using the R Project for Statistical Computing, the single raster was cropped by each of the 2510 DEM .tif files into individual vegetation .tif files. This ensured a one-to-one spatial match of each terrain and vegetation grid cell. The vegetation data come from 5 June 2021 which was selected for several reasons. First, by this point, all of the snow has melted in Wisconsin, and plants are actively growing. At the same time, crops have been planted but are not yet fully grown. The idea behind this was to effectively separate

natural vegetation (i.e., prairie and forests) from agriculture. Experiments with vegetation data from later in the growing season did not effectively show the difference between the abundant coniferous forests of northern Wisconsin and the farms commonly found farther south.

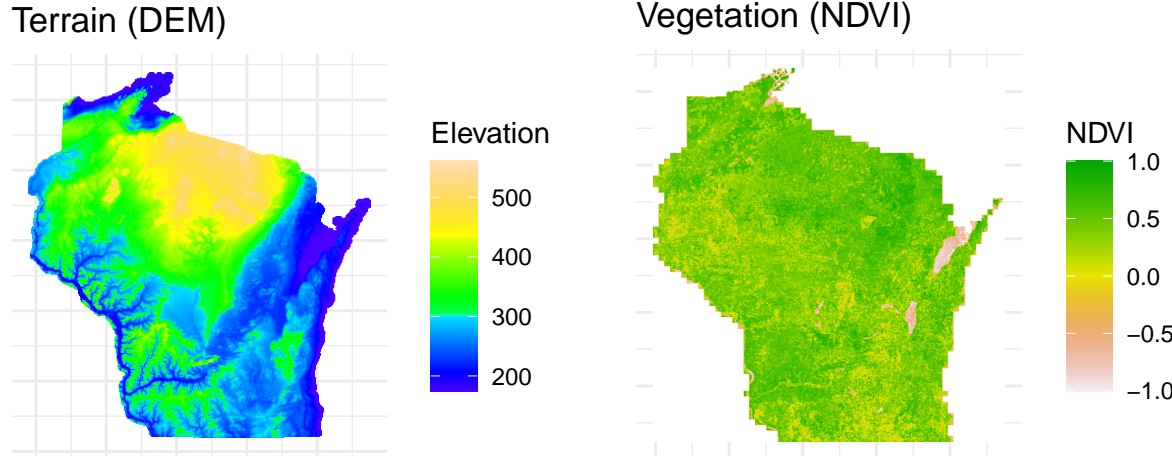

**Figure 1.** Map of terrain (DEM) and vegetation (NDVI) data.

### 2.2. A Priori Modeling

The a priori modeling—which only runs once—effectively serves as a data preparation step before the results are handed over to the web application. The majority of this a priori code was developed with Python 3.7 with a small portion being written in the R Project for Statistical Computing. The major steps for the terrain data involved (1) creating feature vectors using NN models, (2) comparing the feature vectors using cosine distance, and (3) using min-max normalization to effectively scale the results. The three NN models selected are benchmark models in TensorFlow and are commonly used in remote sensing, though other models, such as XCeption and Inception, were tested, but were ultimately not utilized due to their apparent poor performance for the task at hand. Though the authors experimented with applying NN models to the vegetation data similarity, it was ultimately discovered that more direct measures of NDVI, e.g., the mean and standard deviation, better captured similarity as the resolution of individual vegetation images was relatively low, which resulted in the NN models struggling to effectively separate these single-band observations with relatively little structural difference.

#### Model Metrics

In order to create feature vectors, each DEM dataset, stored as a .tif, was first read as a numpy array and resized appropriately based on the required input dimensions of each model. This resizing was accomplished with bilinear sampling. Since the DEM data is effectively a singular band containing one variable—elevation—and NN models often work with three bands (i.e., RGB) images, this singular channel was copied two more times to create an $n \times 3$ array. Each array was then processed through each NN model to create a one-dimensional feature vector.

Following this, each pair of feature vectors was compared using the cosine similarity defined as:

$$\text{cos\_sim} = \frac{A \cdot B}{\|A\|\|B\|}$$

where $A$ is one feature vector and $B$ is another. This is effectively a measure of the angle between two model outputs in vector space, computed by dividing their dot products by their magnitudes. This produces a single value for each pair of images.

In order to scale results between 0 and 1, min-max normalization was used:

$$\text{model\_sim} = \frac{x_{i,j} - min(x)}{max(x) - min(x)}$$

where $x_{i,j}$ represents a single pair of similarity results and $x$ represents the aggregate of all pairs. This was separately completed for each of the three NN models, producing variables `resnet50_sim`, `vgg16_sim`, and `nasnet_sim`. These values were each stored in individual numpy arrays.

After this, the vegetation metrics were computed. The within-image mean NDVI and standard deviation NDVI were each computed, and similarities were computed by retrieving the absolute value of the difference between each pair and then subtracting this value from 1:

$$\text{mean\_ndvi\_sim}'_{i,j} = 1 - abs(\text{ndvi\_mean}_i - \text{ndvi\_mean}_j)$$

$$\text{sd\_ndvi\_sim}'_{i,j} = 1 - abs(\text{ndvi\_sd}_i - \text{ndvi\_sd}_j)$$

These were then min-max normalized to create variables `mean_ndvi_sim` and `mean_sd_sim` and were stored as numpy arrays. Distances (variable `dist`) were also calculated between each image pair and stored in a numpy array.

Finally, the results were aggregated into a SQLite database. Here, each row represents a pair of locations and their corresponding similarity metrics, producing a "tall" rather than "wide" dataset. Since there are 2510 locations in the dataset, the number of rows is equal to the square of the number of locations, i.e., 6,300,100. While this approach produces a reasonable amount of data duplication, leveraging a database in this way allows for shorter query times and more efficient memory usage within the web application. The final database size is a manageable ~450 MB.

### 2.3. Ad Hoc Querying

The querying of results occurs behind the scenes in the web application, which was created with R's web framework, Shiny [26]. On the application's landing page, users are given several input options:

- Exclusion radius in miles (variable `dist`, values: 0–300): following Tobler's First Law of Geography, it was expected that nearby locations would be highly similar and that users may want to exclude options within a certain distance in order to retrieve results from farther away. The default is 0, meaning that no locations are excluded due to nearness.
- Number of similar locations to retrieve (variable `k`, values: 1–10): The default is 5.
- Terrain model (variable `resnet50_sim`, `vgg16_sim`, or `nasnet_sim`, depending on using input from options "ResNet-50", "VGG16", and "NasNet"): The neural network model to use in comparing results.
- Criteria weight for terrain (variable `terrain_scale`, values of 0–1): Relative weight to use for terrain (default of 0.8). This gives end-users flexibility by allowing them to place more or less emphasis on terrain versus vegetation.
- Criteria weight for NDVI mean vs. NDVI standard deviation (variable `veg_mean_scale`, values 0–1): Relative weight to use for each of the two NDVI variables (default of 1). This allows users to place more or less emphasis on total vegetation (i.e., mean NDVI similarity) versus the amount of NDVI variability (i.e., NDVI standard deviation).

Using these input values with the similarities stored in the database, a "total similarity" metric is computed on-the-fly after a user selects input options and clicks on the "Find Similar Landscapes" button:

$$\text{total\_sim} = (\text{terrain\_scale}_u * \text{model\_sim}) +$$

$$(\text{veg\_scale}_u * ((\text{veg\_mean\_scale}_u * \text{veg\_mean\_sim}) + (\text{veg\_sd\_scale}_u * \text{veg\_sd\_sim})))$$

where

$$veg\_scale_u = 1 - terrain\_scale_u$$

and

$$veg\_mean\_scale_u = 1 - veg\_sd\_scale_u$$

Here, variables noted with the subscript "*u*" are either taken from or calculated by user input, whereas the others have been computed a priori and are stored on disk. Effectively, total_sim takes the similarity results and scales them based on the user's desired parameters. This metric represents the combined similarity of terrain and vegetation, enabling users to tailor emphasis on one landscape characteristic or the other to suit a specific use case. The relative weight to place on terrain (terrain_scale$_u$) is multiplied by the terrain similarity scores as computed by the NN models and the cosine distance between the feature vectors (model_sim). Similarly, the weight to place on vegetation (veg_scale$_u$)—which is the additive inverse of the weight placed on terrain—is multiplied by the vegetation similarity results. However, since vegetation similarity considers both NDVI mean similarity (veg_mean_sim) and NDVI standard deviation similarity (veg_sd_sim), the weight to place on each of these vegetation metrics is considered as a part of the larger weight placed on vegetation similarity through the inputs veg_mean_scale$_u$ and veg_sd_scale$_u$, respectively. The metric total_sim could be thought of as simply a weighted average of similarity results scaled by user input options.

Other variables are retrieved from user input and queried from the SQLite database using R's dbplyr package [27] (see Figure 2 for a visual representation of the model). Queries are accomplished quickly due to dbplyr's ability to query databases on disk rather than loading an entire dataset into memory; though users may notice a delay of several seconds, the web application currently operates with only 1 GB of memory and a single CPU core.

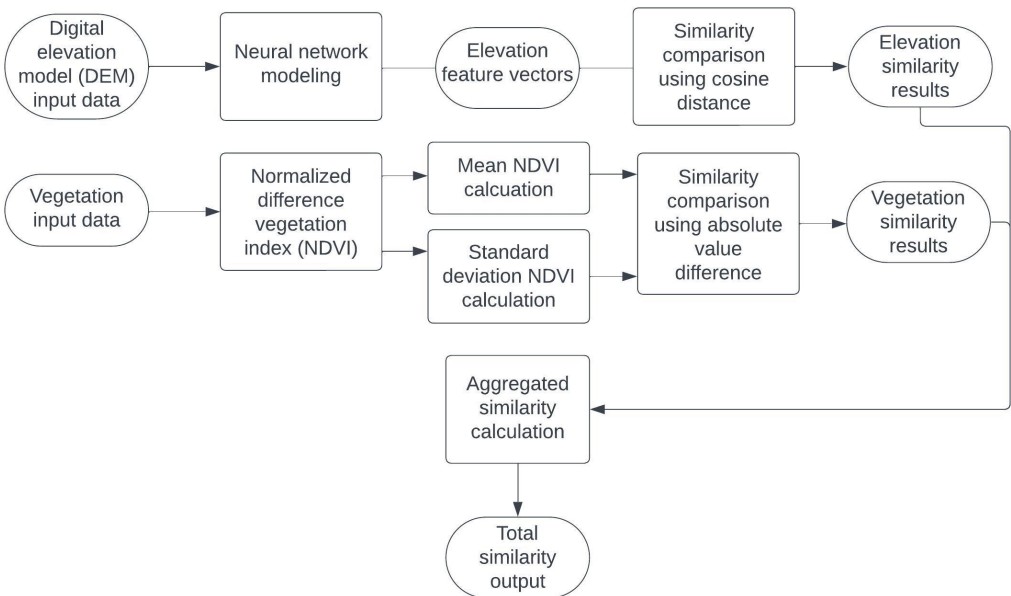

**Figure 2.** Similarity calculation flowchart.

## 3. Case Studies

### 3.1. Observations of General Patterns

In order to better understand how similarity scores are distributed and what the results mean, the authors aggregated the similarity scores of each location pair for every model. Then, the distributions and correlations between measures were investigated. In general,

the aggregated similarity scores produced by ResNet-50 and NasNet are highly left-skewed, with NasNet scores being more leptokurtic (Figure 3). This means that these scores are generally closer to a value of 1, or deemed more similar on average. The VGG16 similarity scores, on the other hand, are far more mesokurtic and slightly right-skewed. This means that, for any given pair of landscapes, the ResNet-50 and NasNet scores are more likely to be scored as more similar, though it should be kept in mind that this is simply a function of how the models produce and compare feature vectors. Min-max normalization helps in compensating for non-normality, but, in the end, such transformations do not alter the ordering of similar images, only the way in which they are represented. The vegetation similarity scores are also left-skewed (Figure 4).

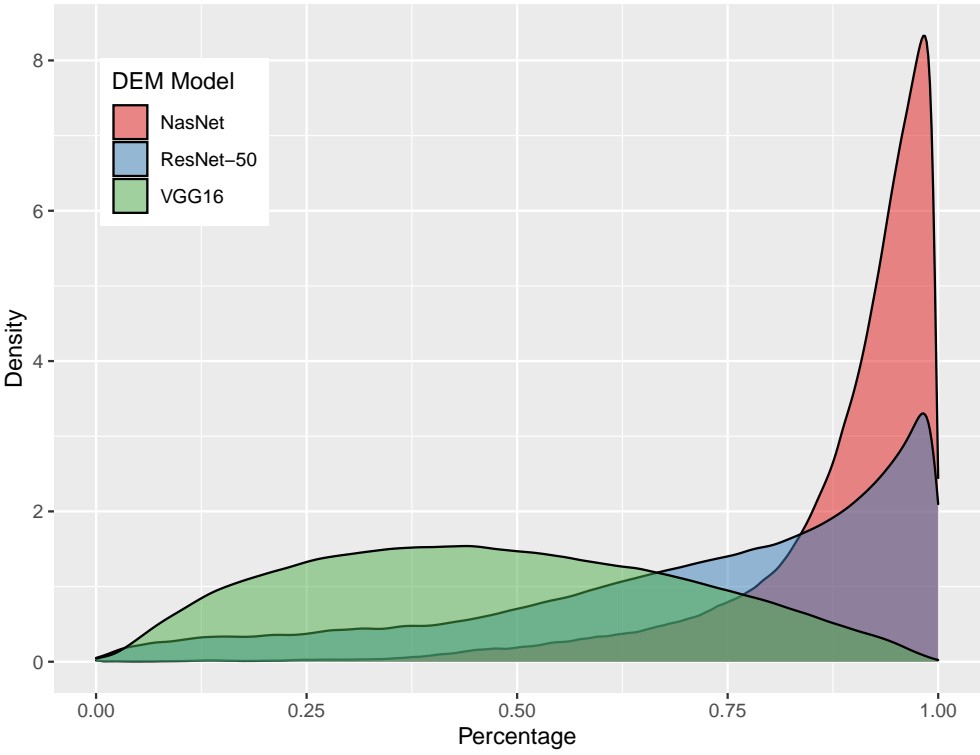

**Figure 3.** Density plots of neural network model similarities.

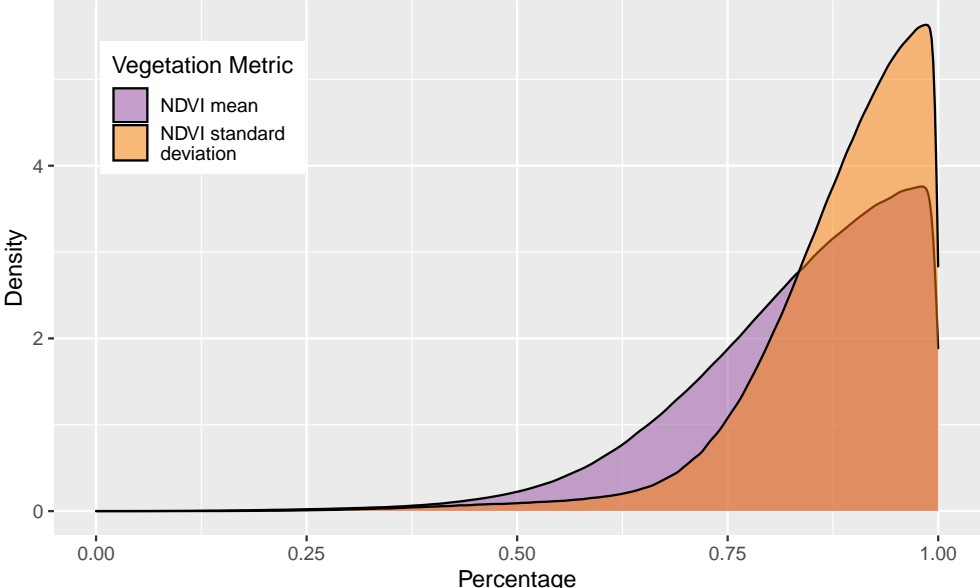

**Figure 4.** Density plots of NDVI similarities.

A correlogram of all variable pairs helps determine how similar model metrics are in terms of what they fundamentally measure (Figure 5). Pairs with stronger correlations exhibit a higher degree of overlap, while those with weaker correlations manifest distinctive measurements. The inclusion of distance in correlation computations also provides insights into spatial dependence. Spearman's $\rho$ is used due to the non-normal nature of the distributions. Correlations among the variables are generally weak with the exception of the terrain variable pairs:

- `resnet50_sim` with `vgg_16_sim`
- `resnet50_sim` with `nasnet_sim`
- `vgg16_sim` with `nasnet_sim`

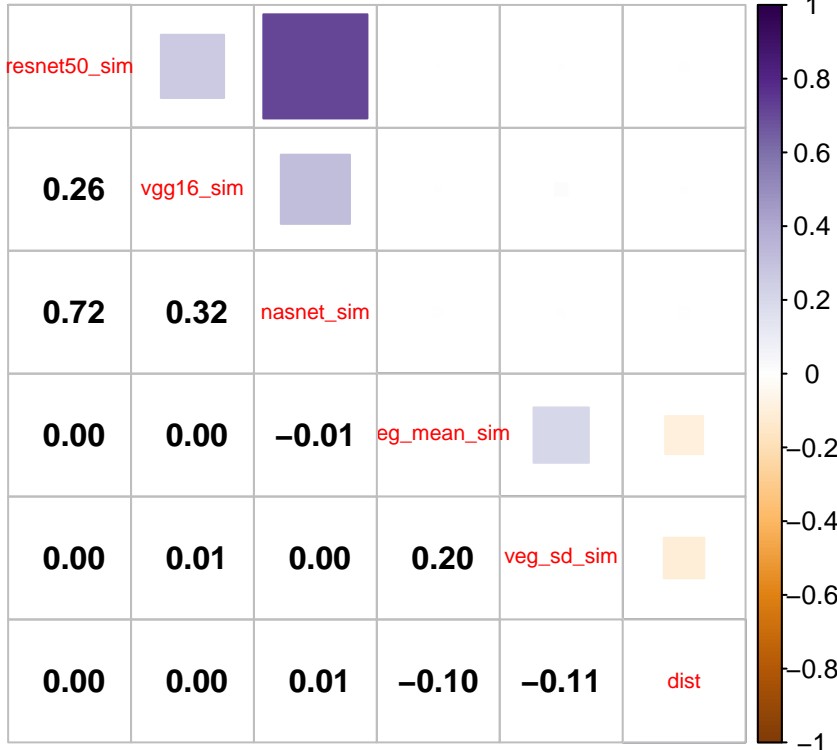

**Figure 5.** Correlogram of similarity results.

The strongest relationship is between `resnet50_sim` and `nasnet_sim` ($\rho = 0.72$), meaning that these variables capture similar things. In turn, this means that `vgg16_sim` is capturing something relatively unique. Despite the high correlations between ResNet-50 and NasNet, we keep both due to the exploratory nature of the web application. Indeed, in practice, the two do seem to function differently.

There is virtually no correlation between the individual vegetation metrics with any of the terrain metrics. On the surface, this appears counterintuitive as the amount of vegetation in a location is, to a certain degree, dependent on characteristics closely tied to the terrain: lithology, topography, and soil. However, while Wisconsin is far from isotropic, its terrain admittedly does not vary nearly as much as a state like Colorado, which straddles the Rocky Mountains. Following this, very flat locations in the state can have wildly different NDVI values—consider, for example, a location of mostly water and one of flat farmland. Further, given the right skew of most model metrics, yet the low amount of correlation between the terrain and vegetation similarity scores, using these two in tandem to produce the total similarity score is wise, as, importantly, the two combined help separate individual observations.

In general, there is a surprisingly low amount of spatial dependence in the data as evidenced by the small Spearman's $\rho$ correlations of the variable `dist` with others. In fact, the relationship between `dist` and `veg_mean_sim` along with the relationship between `dist`

and `veg_sd_sim` are both negative, meaning that nearby locations are likely to be dissimilar in terms of NDVI. While this is a little surprising given the apparent regional differences in Wisconsin with respect to vegetation, the scale of analysis is such that adjacent locations can indeed vary greatly.

### 3.2. Individual Locations

Below, we demonstrate the use of the application with three different locations and parameter configurations. These were chosen intentionally to demonstrate both where the search engine appears to function well and where it does not. Additionally, we retrieve similarity results for three different parts of the state with varied terrain features and vegetation types. We attempt to use a variety of different configuration options, though it is not possible to cover them all.

#### 3.2.1. Location A: Western Wisconsin

This location is located in western Wisconsin, just south of the town of Independence. Situated in the area commonly referred to as the "Driftless Area" due to its lack of evidence for glaciation, it is characterized by relatively steep ridges and dendritic drainage—that is, the terrain appears like branching tree roots (Figure 6). In retrieving similar landscapes, the following model parameters are used:

- `k = 5`
- `terrain_model = 'resnet50'`
- `terrain_scale = 0.8`
- `veg_mean_scale = 1.0`
- `user_dist = 0`

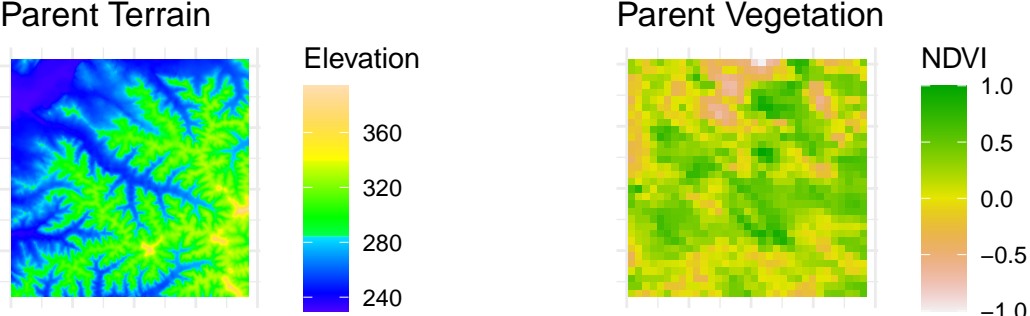

**Figure 6.** DEM and NDVI rasters for Location A (id = 1521).

These are the default options within the web application. So, if a user were to use the application, click on the same location, and obtain results with no modifications, the exact same result would be obtained. With these default options, the majority of the emphasis is placed on the terrain signature—80%—rather than on the vegetation. Additionally, for the 20% of metric emphasis used on vegetation, 100% is used on the total NDVI and none is used on the NDVI variability. No exclusion distance is used in this case, so results may be obtained for locations at any distance away from the parent location (see Figures 7 and 8, and Table 1 for results).

Despite the fact that matched locations are found at varied distances from the parent location—between 5 and 171 miles away—the model appears to work well with this type of location, as matched instances appear very similar, especially those ranked 1, 2, and 5. The dendritic patterns are clearly visible in these matched locations, just like the parent location. The matched location ranked 1 is also located in the Driftless Area, and the matched location ranked 2 is located in the cell adjacent to the parent location, just to the West.

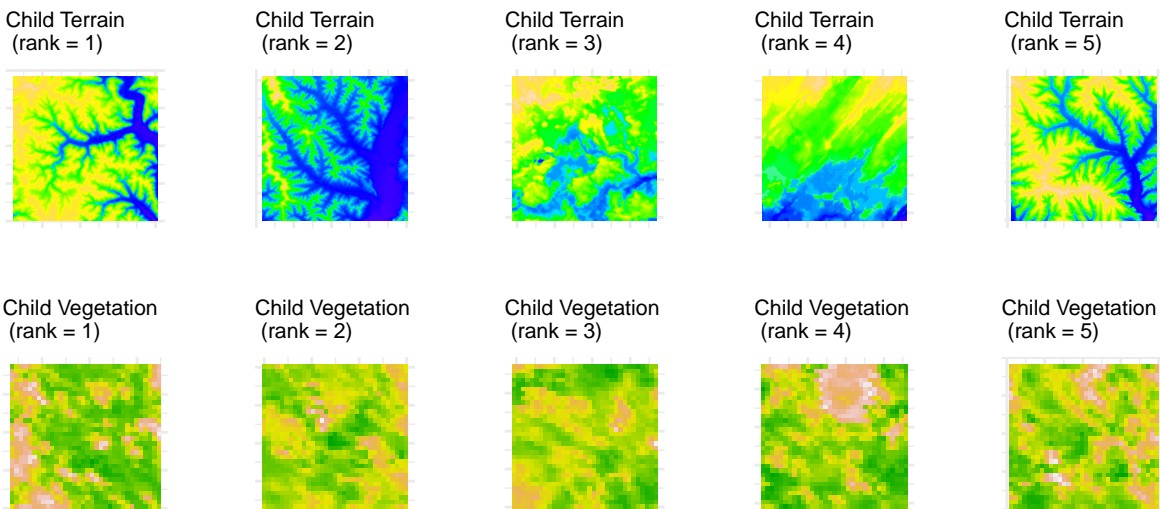

**Figure 7.** Matched locations for Location A (id = 1521).

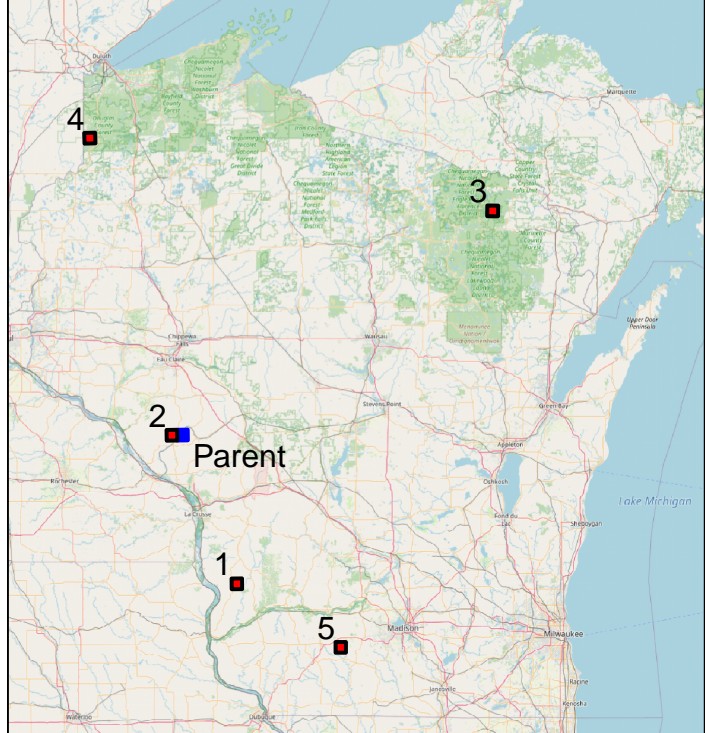

**Figure 8.** Map of matched locations for Location A (id = 1526, shown in blue) labeled by similarity rank (interactive web map available online).

**Table 1.** Similarity results from an example query (id = 1521).

| Similarity Rank | Distance (mi.) | Total Similarity Score | Resnet-50 Similarity | VGG16 Similatiry | Nasnet Similarity | NDVI Mean Similarity | NDVI SD Similatiry | Parent ID | Child ID |
|---|---|---|---|---|---|---|---|---|---|
| 1 | 71 | 0.974 | 0.968 | 0.738 | 0.953 | 0.998 | 0.820 | 1521 | 2073 |
| 2 | 5 | 0.972 | 0.966 | 0.349 | 0.888 | 0.995 | 0.980 | 1521 | 1520 |
| 3 | 171 | 0.971 | 0.970 | 0.613 | 0.923 | 0.973 | 0.986 | 1521 | 450 |
| 4 | 139 | 0.970 | 0.967 | 0.676 | 0.952 | 0.985 | 0.797 | 1521 | 150 |
| 5 | 119 | 0.969 | 0.972 | 0.808 | 0.983 | 0.957 | 0.895 | 1521 | 2289 |

### 3.2.2. Location B: Northern Wisconsin

This location lies in northern Wisconsin in Bayfield County, between the towns of Hayward and Ashland. It is within the Bibon Swamp State Natural Area, and the region is characterized by glacial moraines and a plethora of lakes. While the parent DEM appears to possess significant amounts of water (Figure 9), being in a swamp, this is only the case at certain times of the year. The following model parameters are used:

- `k = 5`
- `terrain_model = 'vgg16'`
- `terrain_scale = 0.5`
- `veg_mean_scale = 0.9`
- `user_dist = 0`

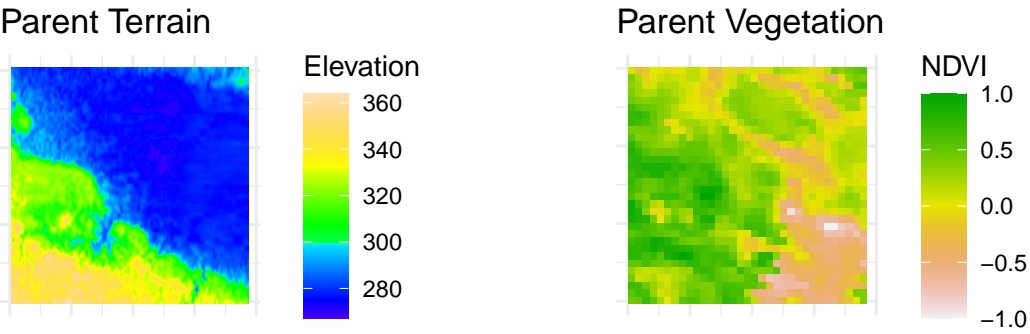

**Figure 9.** DEM and NDVI rasters for Location B (id = 117).

Equal emphasis is placed on terrain and vegetation, and a small amount of emphasis is placed on NDVI variability (10%). Instead of ResNet-50, VGG16 is used, and the exclusion distance is kept at 0 (Figures 10 and 11, and Table 2 for results).

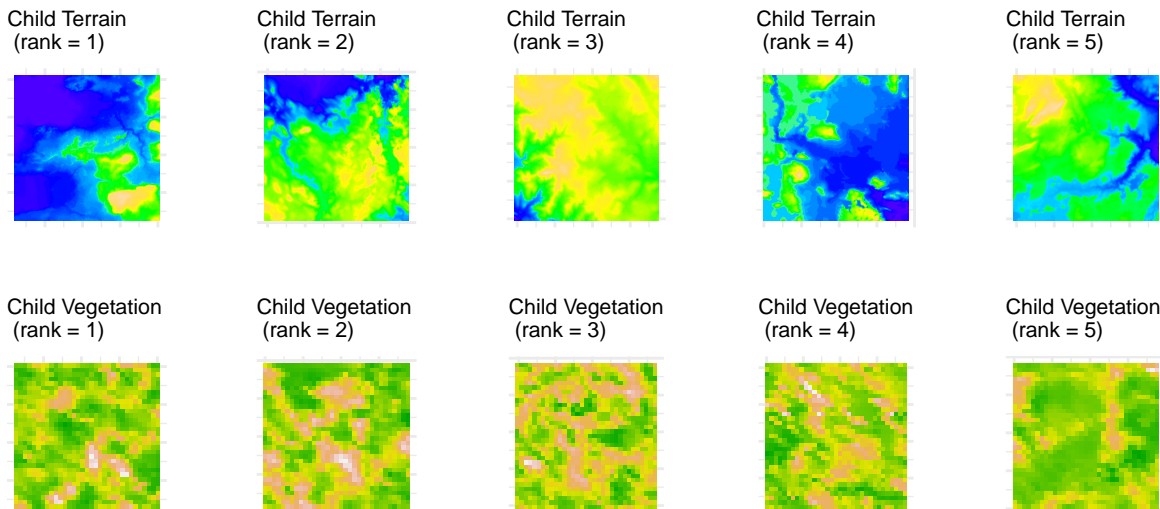

**Figure 10.** Matched locations for Location B (id = 117).

**Table 2.** Similarity results from an example query (id = 117).

| Similarity Rank | Distance (mi.) | Total Similarity Score | Resnet-50 Similarity | VGG16 Similarity | Nasnet Similarity | NDVI Mean Similarity | NDVI SD Similarity | Parent ID | Child ID |
|---|---|---|---|---|---|---|---|---|---|
| 1 | 183 | 0.890 | 0.705 | 0.812 | 0.957 | 0.968 | 0.959 | 117 | 1548 |
| 2 | 226 | 0.884 | 0.937 | 0.811 | 0.974 | 0.959 | 0.942 | 117 | 1772 |
| 3 | 124 | 0.884 | 0.953 | 0.840 | 0.949 | 0.921 | 0.983 | 117 | 1286 |
| 4 | 194 | 0.881 | 0.695 | 0.785 | 0.948 | 0.978 | 0.965 | 117 | 1797 |
| 5 | 283 | 0.872 | 0.792 | 0.831 | 0.944 | 0.917 | 0.872 | 117 | 2309 |

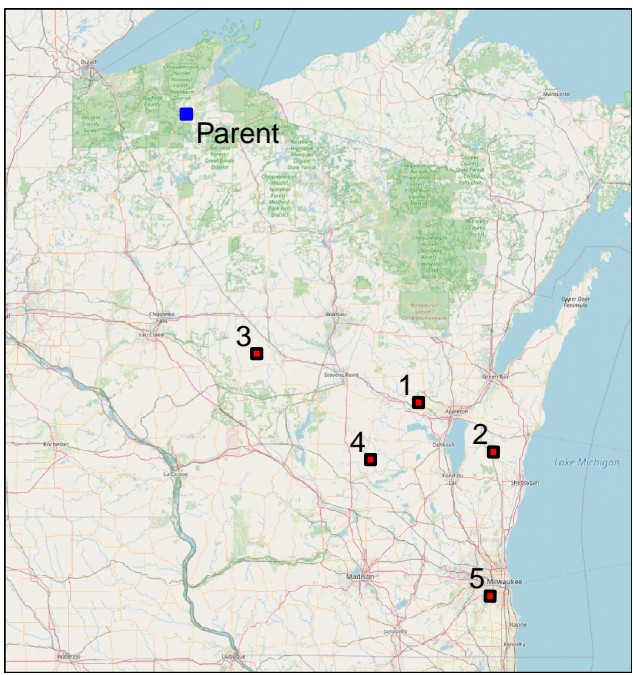

**Figure 11.** Map of matched locations for Location B (id = 117, shown in blue) labeled by similarity rank (interactive web map available online).

Here, all matched locations are relatively distant as the closest matched location is 124 miles away. That said, all matched locations appear materially similar to the parent location, as most are relatively flat and appear to contain significant portions of water.

### 3.2.3. Location C: Urban Milwaukee

This is the only urbanized location evaluated in this paper, and it lies in the southeast part of the state near Lake Michigan. The area is relatively flat with moderately low NDVI values (Figure 12). The following parameters are used:

- k = 5
- terrain_model = 'resnet50'
- terrain_scale = 0.2
- veg_mean_scale = 0.5
- user_dist = 150

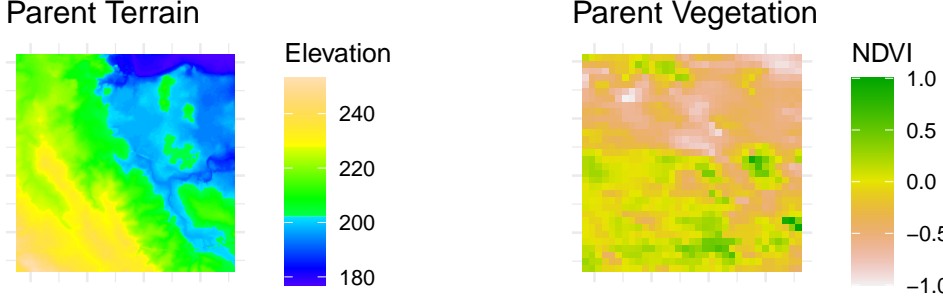

**Figure 12.** DEM and NDVI rasters for Location C (id = 2276).

Here, the influence of terrain is kept small compared to vegetation. Additionally, vegetation variability carries 50% of the overall vegetation influence. This example also makes use of the exclusion distance metric, as all locations within 150 miles of the parent image are excluded from the results. These results have some intriguing facets that are worth discussing (Figures 13 and 14, and Table 3).

First, the terrain images of the matched locations appear visually dissimilar from the parent location, but this is to be expected with only 20% of the overall metric emphasis placed on terrain. The vegetation images appear to be very similar to the parent location, as they have relatively low NDVI values. Notably, all five matched locations appear in a small group; the distances away from the parent location are 262, 253, 241, 213, and 214 miles, respectively, ranked from most similar to least similar. While these are not urban areas, they are certainly visually similar based on the criteria utilized. It is also notable that the area containing the cluster of matched locations is the one substantial area of native prairie in Wisconsin.

Though this example exhibits the difficulty in identifying urban areas as similar to other urban areas, land use is not necessarily dependent upon terrain. Further, at the time of year of this NDVI data—early June—vegetation is less dependent on land use for built-up land than later in the growing season. The inclusion of additional datasets, such as true color aerial photographs, or simply using NDVI from a different time of year, would likely change this result.

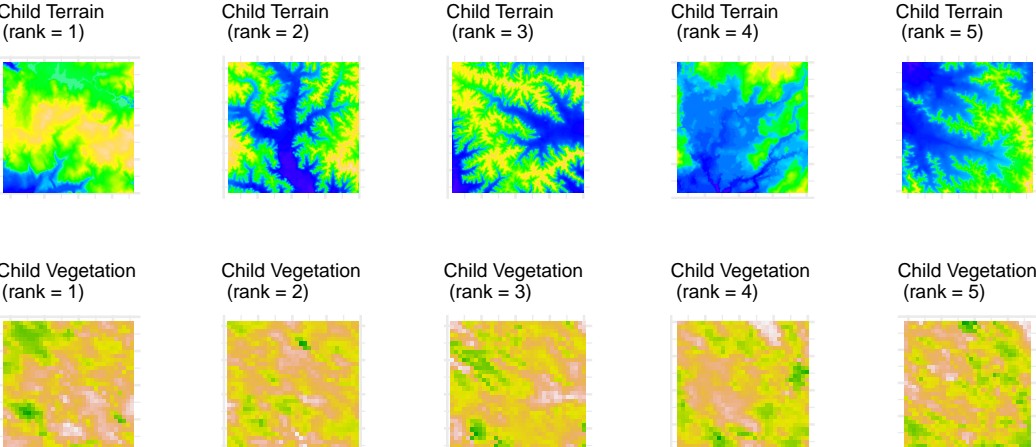

**Figure 13.** Matched locations for Location C (id = 2276).

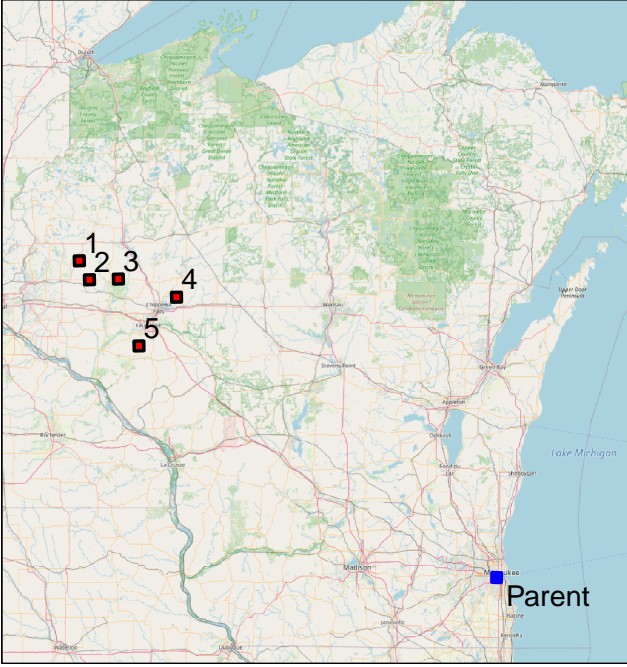

**Figure 14.** Map of matched locations for Location C (id = 2276, shown in blue) labeled by similarity rank (interactive web map available online).

**Table 3.** Similarity results from an example query (id = 2276).

| Similarity Rank | Distance (mi.) | Total Similarity Score | Resnet-50 Similarity | VGG16 Similarity | Nasnet Similarity | NDVI Mean Similarity | NDVI SD Similarity | Parent ID | Child ID |
|---|---|---|---|---|---|---|---|---|---|
| 1 | 262 | 0.982 | 0.964 | 0.676 | 0.785 | 0.995 | 0.978 | 2276 | 767 |
| 2 | 253 | 0.962 | 0.971 | 0.415 | 0.838 | 0.994 | 0.927 | 2276 | 873 |
| 3 | 241 | 0.962 | 0.916 | 0.194 | 0.921 | 0.971 | 0.975 | 2276 | 876 |
| 4 | 213 | 0.954 | 0.861 | 0.428 | 0.781 | 0.961 | 0.995 | 2276 | 996 |
| 5 | 214 | 0.954 | 0.892 | 0.508 | 0.942 | 0.943 | 0.997 | 2276 | 1274 |

## 4. Discussion

Overall, in the authors' experiments, it appears as though VGG16 and ResNet-50 work best for retrieving similar landscapes. Despite the high degree of correlation between the ResNet-50 and NasNet similarity scores, ResNet-50 nevertheless appears to work better. Due to the near infinite number of potential parameter combinations, it is not practical to demonstrate the application using every configuration option and not even with every NN model. The authors leave this further exploration up to the reader. The development of this introductory tool provides a meaningful first step in the domain of NN-based landscape search engines, but, despite the application's utility, its approach is not without drawbacks. Future implementations could improve upon LSE Wisconsin in a variety of ways, yet many of the limitations point to need for a robust, multi-input custom NN architecture designed specifically for landscapes. The subsequent discussion echos this point.

First, higher resolution data encompassing smaller areas may allow for more tangible applications, especially given the low amount of spatial dependence in the data. Lidar-derived 1 m DEM, for instance, could be used in place of the terrain data utilized in LSE Wisconsin. This would, however, increase the end dataset size by a factor of 90, placing considerable strain on a web server equipped with 1GB of memory at the time of writing. Second, one of the most salient limitations is that all models appear, at least to some degree, to struggle in comparing locations covered by large amounts of water. The inclusion of water as a discrete variable in a multi-input NN model would appear prudent, but there can be considerable variation in where water is actually present throughout Wisconsin, particularly in its wide-ranging marshes in the northern part of the state.

Related to this, vegetation data from multiple time periods would allow for different types of comparisons. For instance, giving users the option to select a time period later in the summer may help differentiate urban areas from agricultural land use better, as elucidated in the Milwaukee example. Indeed, such issues would be resolved by the use of a custom NN architecture with several inputs—e.g., terrain, multiple vegetation datasets, land use, aerial photography, and others—but such an approach is inhibited by the inherent subjectivity of "similarity", not to mention a lack of remote sensing test datasets for such problems. Survey-based research would be beneficial in quantifying the degree of landscape likeness. An approach such as the one implemented by Wang et al. [28] with remote sensing scientists would be useful; appropriately ranking a set of images could be used as a test dataset for a custom NN architecture.

Other more obvious extensions include applying this approach to other US states, other locations entirely, or expanding the approach to include an entire country. Such a foray would be ambitious, however, given the necessity of using large, potentially disparate datasets outside of Landsat-derived products. Another ambitious improvement would be in giving users the ability to input their own terrain and/or vegetation datasets for evaluation, though this would require feature vector comparison on-the-fly. Moreover, the increasing availability of user-derived datasets using unmanned aerial systems (UAS) presents opportunities also worth considering for additional improvements. Future work is needed by domain experts to help fine-tune LSE Wisconsin for real use cases and to direct future development.

## 5. Conclusions

This paper introduced a methodology for constructing a neural-network-based landscape search engine and presented a corresponding web application. This is the first tool of its kind for the U.S. state of Wisconsin, and, to the authors' knowledge, it is the first landscape search engine tool that uses NN for landscape search. Through this paper, the authors have demonstrated that benchmark NN models can indeed work for image retrieval with landscape data, and VGG16 and ResNet-50 appear to be the most promising models. Despite the models struggling in locations with significant amounts of water, as it stands now, LSE Wisconsin could nevertheless be used directly for location analysis applications.

This tool marks an important step in the application of image retrieval on remotely sensed datasets, and additional domain applications are likely to emerge with time. Further, the authors hope that LSE Wisconsin ultimately pushes the research community toward a more robust, multi-input landscape search engine tool in the future.

**Author Contributions:** M.H. was the project lead. He created the web application, aided in a priori modeling, and participated in writing the paper; M.D. took the lead on a priori modeling and neural network conceptualization. He also participated in writing the paper; G.A.O.-M. contributed to the remote sensing conceptual framework, application testing, and writing of the paper; P.F.R. contributed to the remote sensing conceptual framework, application testing, and writing of the paper. All authors have read and agreed to the published version of the manuscript.

**Funding:** This project was supported by the Office of Research and Sponsored Programs at the University of Wisconsin-Eau Claire through its Summer Research Experiences for Undergraduates Program.

**Data Availability Statement:** The data used in this project is available at https://gitlab.com/mhaffner/lse-wi/data.

**Acknowledgments:** The authors would like to thank the three anonymous reviewers for their helpful comments and suggestions.

**Conflicts of Interest:** The authors declare no conflict of interest. The founding sponsors had no role in the design of the study; in the collection, analyses, or interpretation of data; in the writing of the manuscript; or in the decision to publish the results.

**Sample Availability:** The web application is available at https://uwec-geog.shinyapps.io/lse-wi/ (accessed on 6 August 2023). The source code for the web application is available at https://gitlab.com/mhaffner/lse-wi. The source code for the data extraction and a priori modeling is available at https://gitlab.com/mhaffner/landscape-search-engine.

## Abbreviations

The following abbreviations are used in this manuscript:

| | |
|---|---|
| CNN | Convolutional Neural Network |
| DEM | Digital Elevation Model |
| NDVI | Normalized Difference Vegetation Index |
| NN | Neural Network |
| NN | NasNet |
| DL | Deep Learning |
| USGS | United States Geological Survey |
| DNR | Department of Natural Resources |
| GDAL | Geospatial Data Abstraction Library |
| GIScience | Geographic Information Science |
| NASA | National Air and Space Administration |
| MODIS | Moderate Resolution Imaging Spectrometer |
| VGG | Visual Geometry Group |

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
