# Peer review of "A Neural-Network-Based Landscape Search Engine: LSE Wisconsin"

_applsci, doi:10.3390/app13169264_

Round 1

Reviewer 1 Report

The authors presented of a web-based, landscape search engine for the US state of Wisconsin. The application allows to user  selecting a location on the map and find similar locations based on terrain and vegetation characteristics. They used three neural network models on digital elevation model data, comparing vegetation data via NDVI mean and standard deviation, demonstrating that VGG16 and ResNet50 expose more favorable results. The authors think that the tool could be important first step for performing a more robust, multi-input, high resolution landscape search engine.

Comments:

1)     In opinion of this reviewer, the authors should present explicitly their principal scientific contributions, Also, they should compare their designed tool with similar ones, The authors presented in brief form several criteria used in calculating similarity type metrics, but they do not explain details in this part. Such description does not permit better understanding of their tool.

2)     The authors should explain for better understanding by potential reader how they use standard metrics: cos_sim, model_sim. Please define parameters A, B, and x from images used in equations (pag. 4, lines 141-143). This reviewer considers that that same arrangement should be performed in following equations (page 4, lines 146-150). It is difficult for potential reader understand the explications presented in this part.

3)     In page 5 (lines 176-182), the authors presented equation for computing “total similarity” metric. For a reader, it would be difficult understand the reason of this parameter, as well as their justification.. The author never explained any details in this part.

4)     This reviewer recommends introducing additional subsection “Discussion” where the authors can put several sentences exposed in Conclusion section, presenting in the conclusion only principal results, limitations of the proposed tool, and future research that can increase performance of authors´ approach.

Author Response

Many thanks for your helpful comments. The organizational suggestions were particularly beneficial.

"In opinion of this reviewer, the authors should present explicitly their principal scientific contributions, Also, they should compare their designed tool with similar ones, The authors presented in brief form several criteria used in calculating similarity type metrics, but they do not explain details in this part. Such description does not permit better understanding of their tool."

We modified the introduction to state the principal scientific contributions of the paper (p. 2, lines 32-38). We added explanations to how the existing landscape search tools work along with their deficiencies (p. 2, lines 62-66), and we greatly expanded explanations on how our similarity metrics work (p. 4, lines 153-155; p. 5-6, lines 189-203).

"The authors should explain for better understanding by potential reader how they use standard metrics: cos_sim, model_sim. Please define parameters A, B, and x from images used in equations (pag. 4, lines 141-143). This reviewer considers that that same arrangement should be performed in following equations (page 4, lines 146-150). It is difficult for potential reader understand the explications presented in this part."

We added further explanations to the equations cos_sim and model_sim (p. 4, lines 153-155; p. 5-6, lines 189-203).

"In page 5 (lines 176-182), the authors presented equation for computing “total similarity” metric. For a reader, it would be difficult understand the reason of this parameter, as well as their justification. The author never explained any details in this part."

We added both further explanation and motivation for the total similarity metric (p. 5-6, lines 189-203). We also added a flowchart which explains how the similarity metrics are calculated (p. 6).

"This reviewer recommends introducing additional subsection “Discussion” where the authors can put several sentences exposed in Conclusion section, presenting in the conclusion only principal results, limitations of the proposed tool, and future research that can increase performance of authors´ approach."

We added a Discussion section and moved most of the content previously in the Conclusion here. Further, we used the conclusion section to highlight the principal results, limitations, and future research directions (p. 15-16, lines 325-374).

Reviewer 2 Report

This is an interesting paper. There are some suggestions for revision.

1. The motivation is not clear. Please specify the importance of this paper.

2. Please highlight the novelty of this paper in introduction.

3. Please discuss more recently published solutions, espcially the solutions published in 2023 and 2022.

4. More technical details should be given.

5. Please discuss whether the solution can be applied to different locations.

6. It is better to add comparative experiments. 

7. Please specify the special features of the dataset used in this paper. 

NA

Author Response

Thank you for your helpful comments. Upon reading them, we realized that we did indeed need to explain the calculations better and have done so (along with updating the paper with your other suggestions).

"The motivation is not clear. Please specify the importance of this paper."

We modified the last sentence of the first paragraph to make the motivation more clear (p. 1, lines 23-24).

"Please highlight the novelty of this paper in introduction."

We added several sentences to the end of the introduction which highlight the most important contributions (p. 2, lines 32-38).

"Please discuss more recently published solutions, especially the solutions published in 2023 and 2022."

We added a number of studies to the literature review with a focus on more recent studies (p. 2, lines 40-46, 80; p. 3, lines 82-84).

"More technical details should be given."

In line with comments from another reviewer, we added more details about the cosine distance calculation (p. 4, lines 149-152), the total similarity metric, how these are calculated, and the motivation for calculating them in this way (p. 4, lines 153-155; p. 5-6, lines 189-203). We also included a flow chart which explains how the similarity metric is calculated (p. 6).

"Please discuss whether the solution can be applied to different locations."

This was discussed in the Conclusion section (now moved to the Discussion; p. 15, lines 354-355)

It is better to add comparative experiments.

"This would certainly be useful, but two of the cited studies which created landscape search toolkits do not have publicly available tools. We added a sentence on this to clarify their lack of availability (p. 2, lines 65-66). Further, comparisons are also somewhat precluded by the difference in fundamental data sources in these other implementations, making direct comparisons challenging. This, combined with the paper's present length at 17 pages makes additional comparisons beyond the scope of this study."

"Please specify the special features of the dataset used in this paper."

We modified the first sentence of the subsection "Data Extraction" to clarify the uniqueness of the data sources. The following two paragraphs describe them in more detail (p. 3, lines 104-127).  

Reviewer 3 Report

The authors have to introduce the subject clearly. The LR must be expanded and added more related studies.

The method is not straightforward, I recommend adding a flow graph at the beginning of the method section

What is the benefit of studying the similarities between features, I think that is indicated by the correlation. That may be abstracted using correlation functions. 

Your results need more clarification 

Why did you just use these mentioned CNNs

Use formal expressions for equations.

The discussion part has been compared with existing studies.

The conclusion has mentioned the proposed study's benefits, how can we improve it in the future, and where it can be used.

Author Response

Thank you for your helpful comments and recognition of the paper's strengths. The suggestion about the flow graph was particularly helpful.

"The authors have to introduce the subject clearly. The LR must be expanded and added more related studies."

We added a number of studies to the literature review with a focus on more recent studies (p. 2, lines 40-46, 80; p. 3, lines 82-84). Additionally, we modified the introduction to note the project's motivation (p. 1, lines 23-24) and highlighted the principal contributions of the project (p. 2, lines 32-38).

"The method is not straightforward, I recommend adding a flow graph at the beginning of the method section."

We added a flow chart to the manuscript (Figure 2, p. 6).

"What is the benefit of studying the similarities between features, I think that is indicated by the correlation. That may be abstracted using correlation functions."

We added sentences clarifying this in the Case Studies section (p. 7, lines 222-227).

"Your results need more clarification"

In line with comments from another reviewer, we added more details about the cosine distance calculation, the total similarity metric, how these are calculated, and the motivation for calculating them in this way (p. 4, lines 153-155; p. 5-6, lines 189-203).

"Why did you just use these mentioned CNNs"

We added a sentence to clarify this in the subsection "A Priori Modeling" (p. 4, lines 134-136).

"Use formal expressions for equations."

We used MathJax to produce equations and used commonly accepted notations where applicable. Due to the experimental nature of this study, however, we opted to use descriptive variable names rather than short, nondescipt letters or symbols in order to make the paper more readable. That said, we did modify equations slightly to conform with another reviewer's suggestions which should make equations more easier to understand (p. 4-6).

"The discussion part has been compared with existing studies."

We added additional content on existing landscape search tools to the Background section (p. 2, lines 63-64)

"The conclusion has mentioned the proposed study's benefits, how can we improve it in the future, and where it can be used."

In line with another reviewer's suggestions, we moved most of this content to a new Discussion section. 

Round 2

Reviewer 1 Report

Normal practice in presenting revised manuscript includes color marking the parts of the text that have been changed. The authors did not use such a practice, generating difficulties in understanding changes done in novel version. The authors increased the quality of their manuscript, presenting brief explications into the text of this study. In opinion of this reviewer by now, it is difficult for a reader in understanding the reason to use such definition for “total similarity” metric. Also, the authors presenting the lines marked as changed ones should be careful because some of these lines are the same as in previous version.

Author Response

"Normal practice in presenting revised manuscript includes color marking the parts of the text that have been changed. The authors did not use such a practice, generating difficulties in understanding changes done in novel version."

Our apologies for not following this practice and making it difficult to understand the updates in the new manuscript. We had assumed the common practice now was to utilize a "git diff" on the first and second .tex files in order to see manuscript changes. We have attached a plain text version of the diffed changes from the first and second drafts for your convenience. Additionally, since the changes between the second and third drafts are minor, we marked these additions in red.

"The authors increased the quality of their manuscript, presenting brief explications into the text of this study."

Thank you for your positive remarks here.

"In opinion of this reviewer by now, it is difficult for a reader in understanding the reason to use such definition for “total similarity” metric."

In addition to the supplementary explanations of the total similarity metric in the second draft, we added another sentence clarifying the role of this metric in this draft (lines 197-199).

"Also, the authors presenting the lines marked as changed ones should be careful because some of these lines are the same as in previous version."

Our apologies for this oversight. New changes are marked in red in this draft.

Reviewer 2 Report

All my concerns have been addressed. I recommend this paper for publication.

Author Response

"All my concerns have been addressed. I recommend this paper for publication."

Thank you for your helpful comments and suggestions!

Reviewer 3 Report

the authors improved the paper 

Author Response

"the authors improved the paper "

Thank you for your helpful comments and suggestions!